# Discovery of a heparan sulfate binding domain in monkeypox virus H3 as an anti-poxviral drug target combining AI and MD simulations

**Bin Zheng[1†], Meimei Duan[2†], Yifen Huang[1†], Shangchen Wang[1†], Jun Qiu[1], Zhuojian Lu[1], Lichao Liu[1], Guojin Tang[1], Lin Cheng[2*], Peng Zheng[1,3*]**

[1]State Key Laboratory of Coordination Chemistry, Chemistry and Biomedicine Innovation Center (ChemBIC), School of Chemistry and Chemical Engineering, Nanjing University, Nanjing, China; [2]Institute for Hepatology, National Clinical Research Center for Infectious Disease, Shenzhen Third People's Hospital, Shenzhen, China; [3]Nanjing Drum Tower Hospital, Affiliated Hospital of Medical School, Nanjing University, Nanjing, China

**\*For correspondence:**
chenglin252@163.com (LC);
pengz@nju.edu.cn (PZ)

†These authors contributed equally to this work

**Competing interest:** The authors declare that no competing interests exist.

## eLife Assessment

This work presents **important** findings regarding the interaction of the monkeypox virus (MPXV) attachment H3 protein with the cellular receptor heparan sulfate and the use of this information to develop antivirals potentially effective against all orthopoxviruses. Using a combination of state-of-the art computational and wet experiments the authors present **convincing** evidence to sustain their claims. These results will interest those working on basic orthopoxviruses biology and antiviral development.

**Abstract** Viral adhesion to host cells is a critical step in infection for many viruses, including monkeypox virus (MPXV). In MPXV, the H3 protein mediates viral adhesion through its interaction with heparan sulfate (HS), yet the structural details of this interaction have remained elusive. Using AI-based structural prediction tools and molecular dynamics (MD) simulations, we identified a novel, positively charged α-helical domain in H3 that is essential for HS binding. This conserved domain, found across *orthopoxviruses*, was experimentally validated and shown to be critical for viral adhesion, making it an ideal target for antiviral drug development. Targeting this domain, we designed a protein inhibitor, which disrupted the H3-HS interaction, inhibited viral infection in vitro and viral replication in vivo, offering a promising antiviral candidate. Our findings reveal a novel therapeutic target of MPXV, demonstrating the potential of combination of AI-driven methods and MD simulations to accelerate antiviral drug discovery.

## Introduction

The mpox virus (also known as monkeypox virus), a zoonotic pathogen within the *Orthopoxvirus* genus, has emerged as a global health concern. This genus also includes the variola virus (VARV), the causative agent of smallpox, and vaccinia virus (VACV), which has been used as a live vaccine against smallpox (*Lum et al., 2022*; *The Lancet Infectious, 2022*; *Figure 1A*). Historically endemic to Central and West Africa, MPXV gained international attention during the global outbreaks in 2022 (clade

**Figure 1.** Structural analysis of the H3 protein in monkeypox virus (MPXV). (**A**) Phylogenetic tree depicts the evolutionary relationships of H3 within the Poxviridae family, highlighting MPXV (blue circle), variola virus (VARV) (red circle), and vaccinia virus (VACV) (green circle). (**B**) Schematic of MPXV adhesion to the cell surface. Viral particles bind to cell surface via specific interaction such as between adhesion protein H3 and heparan sulfate (HS), followed by membrane fusion mediated by the fusion complex, allowing entry into the host cell. (**C**) The amino acid sequence of MPXV H3 shows the newly discovered helical structure (240–282, highlight in yellow), the Mg(II) (green) binding site, and other potential HS binding motifs (blue underlines). (**D**) AlphaFold2 (AF2) prediction of MPXV H3 structure. The left and right panels show different orientations of the H3 structure (rotated 180°). The blue region corresponds to H3(1–239), which has a homologous crystal structure (VACV H3, PDB code: 5EJ0). The yellow region represents the AF2-predicted structure of H3(240–282), which remains unresolved in the crystal structure of the homologous protein. All potential glycosaminoglycans (GAGs) binding motifs are highlighted with a yellow background. (**E**) Molecular dynamics (MD) simulation snapshot of H3 on a DPPC (dipalmitoylphosphatidylcholine) membrane. H3 is anchored to the membrane through its transmembrane region (residues 283–306, in gray).

The online version of this article includes the following video and figure supplement(s) for figure 1:

**Figure supplement 1.** Structural prediction protein-membrane modeling of monkeypox virus H3 protein.

**Figure 1—video 1.** Molecular dynamics (MD) simulation trajectories of H3 with DPPC (dipalmitoylphosphatidylcholine) membrane.

https://elifesciences.org/articles/100545/figures#fig1video1

---

IIb) and 2024 (clade Ib), leading the World Health Organization (WHO) to declare it a public health emergency of international concern twice within 3 years (*O'Toole et al., 2023*; *Kirby, 2023*; *Vakaniaki et al., 2024*). Although MPXV has a lower fatality rate than smallpox (2–10% versus 30%), the absence of specific antiviral treatments continues to pose significant health risks (*Li et al., 2023*; *Liu et al., 2022*; *Zahmatyar et al., 2023*; *Zaeck et al., 2023*; *Yu et al., 2023*; *Peng et al., 2023*).

Viral adhesion to host cells is a critical initial step in the infection process for many viruses. Proteins that mediate these interactions have been identified as essential therapeutic targets, offering promising opportunities for antiviral therapies. In *orthopoxviruses* such as MPXV, the H3 protein plays a pivotal role in facilitating viral adhesion by binding to cell-surface heparan sulfate (HS) (*Moss, 2020*; *Gray et al., 2019*; *da Fonseca et al., 2000a*), a glycosaminoglycan (GAG) essential for viral entry. H3-specific neutralizing monoclonal antibodies have shown to protective effects for rabbits and immunodeficient mice against lethal poxvirus infections (*Crickard et al., 2012*). And H3 is a key component in recent mpox vaccines (*Zuiani et al., 2024*; *Gulati et al., 2022*). Despite the central role of H3 in MPXV infection and protection, the structural details of its specific interaction with HS remain elusive.

Understanding the molecular mechanisms underlying the H3-HS interaction is critical for advancing antiviral drug development. H3 is highly conserved across *poxviruses* (*Figure 1A*), including VARV and VACV, making it a potential target for broad-spectrum therapies. The recent resurgence of MPXV infections has underscored the urgent need for specific antiviral treatments. Given the conserved nature of H3 across all known MPXV lineages, including clade Ib, which has been linked to the 2024 outbreak, designing therapies that target H3 could offer a robust solution to current and future outbreaks.

However, efforts to resolve the complete structure of H3-HS complex have been hampered by the dynamic and flexible nature of HS and the tendency of H3 self-cleavage. Classic structural biology techniques, such as X-ray crystallography, have been unable to capture the full interaction between H3 and HS due to these complexities. Therefore, alternative approaches are required to fully elucidate this critical interaction.

To address these challenges, we employed advanced AI-based protein structural prediction and generation tools, including AlphaFold2 (AF2) and RFdiffusion, alongside classic molecular dynamics (MD) simulations. These state-of-the-art computational techniques enabled us to identify key HS binding sites within H3 and to characterize the binding mechanisms of the previously uncharacterized α-helical domain (*Jumper et al., 2021*; *Phillips et al., 2005*; *Gräter et al., 2005*; *Kim et al., 2013*; *Brooks et al., 2024*). Corroborated by various experimental methods (*Rief et al., 1997*; *Sieben et al., 2012*; *Raab et al., 1999*; *Müller et al., 2009*), the use of AI-driven tools proved pivotal in overcoming the limitations of traditional structural methods, allowing us to predict and validate the structure-function relationship of the H3-HS interaction. A powerful approach for unraveling complex protein-glycan interactions is demonstrated, offering new pathways for the development of broad-spectrum antiviral therapies (*Dauparas et al., 2022*; *Watson et al., 2023*).

## Results

### Identification of a novel helical domain for HS binding in H3

The H3 protein of MPXV is a 324 amino acid transmembrane protein conserved across all 12 identified *orthopoxviruses*. Previous studies have shown that its extracellular domain (residues 1–282) binds to cell-surface HS (*Singh et al., 2016*; *da Fonseca et al., 2000b*), facilitating viral entry in conjunction with other adhesion proteins and the fusion protein system (*Figure 1B*; *Lin et al., 2000*). Inhibiting the HS binding to H3, therefore, represents a promising strategy for developing broad-spectrum antiviral therapies against *orthopoxviruses*. While most structure of the extracellular region of VACV H3 (residues 1–240; PDB code: 5EJ0) has been solved, the dynamic and heterogeneous nature of HS as a biopolymer complicates the identification of precise binding sites.

To identify HS binding sites on the H3 protein, we first performed a comprehensive sequence and structural analysis. As a highly negatively charged GAGs, it is believed that HS interacts with positive charge residues on proteins by electrostatic interaction. Previous studies have indicated that the positively charged Mg (II) ion in H3 is critical for HS binding. Therefore, we focused on regions rich in basic amino acids—lysine (K), arginine (R), and histidine (H). Given that only the H3 structure from VACV was available (*Watson et al., 2023*), we used AF2 to model the MPXV H3 structure (*Figure 1—figure supplement 1A*). Our analysis revealed three potential HS binding motifs rich in positively charge residues: Motif 1 (K95, R96, R100), Motif 2 (K141, K146, K147), and Motif 3 (K161, K163) (*Morgan and Wang, 2013*), all located within the homologous structure of VACV H3 (*Figure 1C and D*).

Interestingly, we also identified a cluster of seven additional positively charged residues (R242, H247, R248, K253, R259, K266, R267) in the protein C-terminal region. This region had not been previously structurally characterized due to its self-cleavage during sample preparation for X-ray diffraction

studies, but was retained in all other biochemical and viral assays (*Watson et al., 2023*; *Singh et al., 2016*; *da Fonseca et al., 2000b*). AF2 predicted these residues to form a three-helical domain with high confidence (pLDDT score: 82.9 for overall structural; 80.6 for the helical domain, 100 for the maximum score), suggesting it may serve as an HS binding site as a structured domain (*Figure 1D*). We also tested other models, such as ESMFold (pLDDT:70.6; 50.7) and RoseTTAFold2 (pLDDT: 70.2; 50.7) (*Figure 1—figure supplement 1B and C*; *Lin et al., 2023*; *Baek et al., 2024*). They all predict a helical structure. But their results are different in detail and had lower confidence scores. Thus, we used AF2 for further studies. Another concern is that the helical domain near the virus surface. Thus, we modeled the complete structure of H3 in virus using a model DPPC (dipalmitoylphosphatidylcholine) membrane. It showed a dynamic movement between the helical domain and the membrane, with a maximum distance exceeding 1 nm (*Figure 1E*, *Figure 1—figure supplement 1D and E*, *Figure 1—video 1*). Thus, enough space be present for possible HS binding.

Flexible molecular docking and MD simulations identified the exact HS binding sites in the H3 protein (*Eberhardt et al., 2021*; *Lee et al., 2016*). We used a 20-repeat HS unit with a specific composition for detailed analysis ([IdoA2S-GlcNS6S-IdoA-GlcNS(3,6S)]$_5$) (*Figure 2—figure supplement 1*). A 500 ns MD simulation established the equilibrium conformation of H3, and subsequent docking with AutoDock Vina identified four high-scoring conformations for each motif (*Figure 2A*, *Figure 2—figure supplement 2*). The docking score calculated by the software reflects the total energy of interactions between molecules. Lower scores (more negative values) indicate more stable binding between the molecules. To address the dynamic nature of HS, we extended our simulations to three times additional 1000 ns MD simulations for each conformation. RMSD (root mean square deviation) analysis demonstrated notably stable binding for HS to both Motif 1 and the newly identified helical domain (*Figure 2—video 1*), with RMSD values consistently below 4 nm, indicating minimal structural fluctuations during interaction (*Figure 2B*).

Moreover, we quantitatively analyzed the binding free energy between H3 and HS in different conformational states using umbrella sampling techniques. A harmonic potential was applied to HS and gradually pulled away from H3, generating a series of reaction coordinates (*Figure 2C*, *Figure 2—figure supplement 3*, *Figure 2—video 2*). The green force-extension curve represents the force during this stretching process. This curve, along with the corresponding PMF (potential of mean force) variation shown in *Figure 2—figure supplement 3*, demonstrated the free energy changes throughout the umbrella sampling process, revealing that the helical domain possesses the highest binding affinity with a ΔG of –45 kcal/mol (*Figure 2D*).

Previous studies have suggested the critical role of the Mg(II) ion in stabilizing HS binding, where its removal reduces binding efficiency (*Singh et al., 2016*). Indeed, the α-helical domain is very close to the Mg(II)-bound region in H3. Notably, these two parts appear to form a distinct cavity-like binding pocket for HS (*Figure 2E*). Thus, we conducted a 1000 ns REMD (replica exchange molecular dynamics) simulation of the H3-HS complex to further understand the binding dynamics. REMD, an advanced sampling technique, helps overcome the kinetic barriers typically encountered in conformational transitions by utilizing a series of temperature-controlled replicas to enhance the exploration of the conformational landscape (*Sugita and Okamoto, 1999*). For this study, we set up 64 replicas over a temperature range from 310 K to 387 K, cumulatively simulating for 64 μs. This comprehensive simulation captured the HS molecule progressively binding deeper into a cavity formed by the α-helical domain, eventually interacting with the Mg (II) ion as hypothesized (*Figure 2E*, *Figure 2—video 3*).

The free energy landscape, mapped out from these simulations, shows HS transitioning from a stable interaction with the helical domain across an energy barrier into the Mg (II)-enhanced cavity. This dual interaction—first with the helical domain and then with the Mg(II) site—was consistently observed to stabilize the complex further, evidenced by a significantly lower binding free energy of –67 kcal/mol (*Figure 2—figure supplement 3*; *Govind Kumar et al., 2023*). These results underscore the dynamic nature of the H3-HS interaction and validate our model of sequential binding, which could be critical for designing inhibitors that target these specific interactions.

Given the crucial role of electrostatic interactions in the HS-H3 binding process, we conducted an exhaustive analysis of salt bridge formations across all MD simulation trajectories. This analysis focused on the formation of salt bridges over time between the basic amino acids (R, K, H) of H3 and HS, with brighter areas indicating more frequent formation of salt bridges and robust interaction (*Figure 2F*, *Figure 2—figure supplement 4*). A bar graph accompanied these maps, quantifying the average

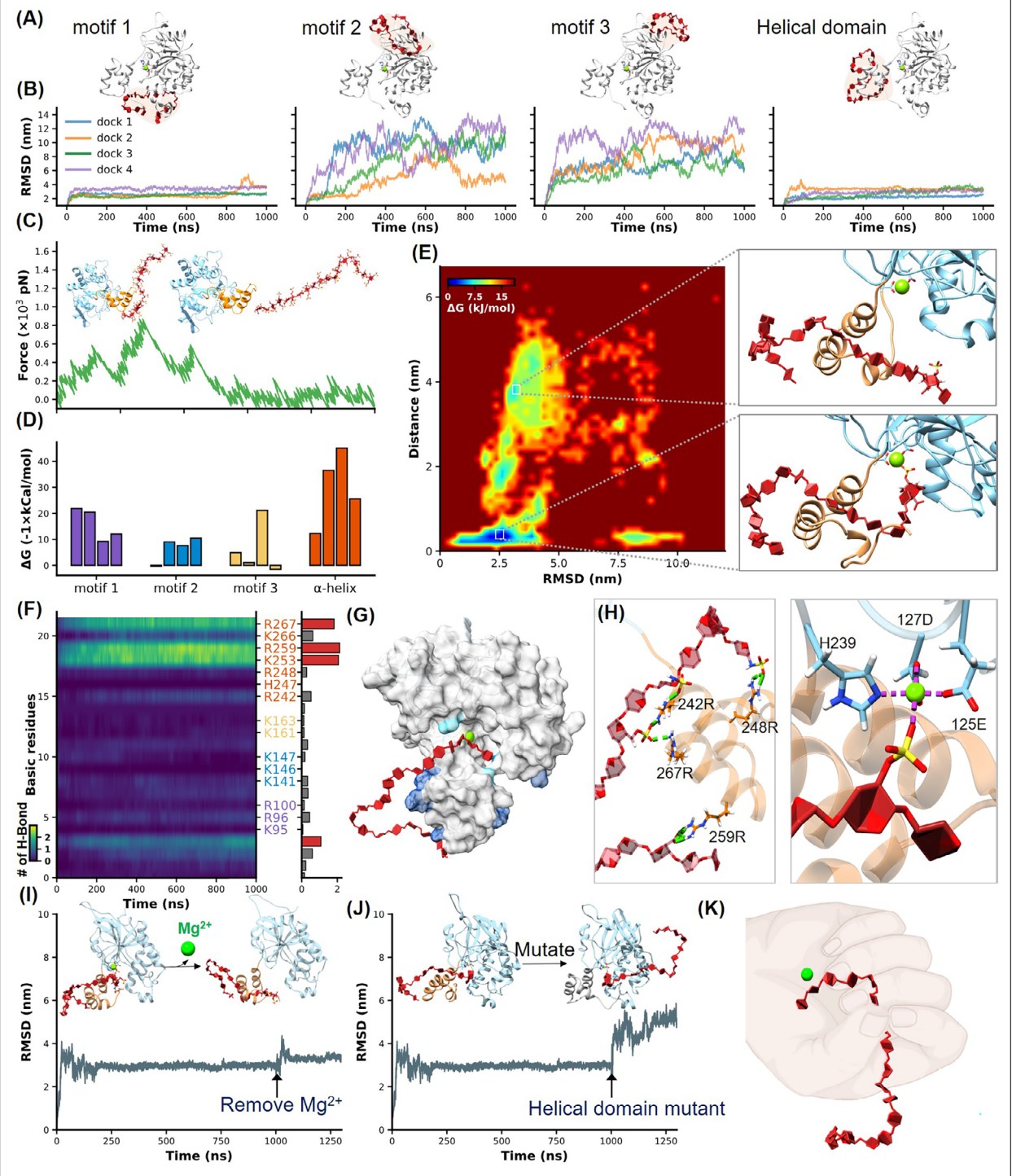

**Figure 2.** Identification of heparan sulfate (HS) binding motifs in H3 by molecular dynamics (MD) simulation and docking. (**A–B**) Cartoons show docking results of HS with Motifs 1, 2, 3, and the helical domain, respectively. (**B**) Panels show RMSD (root mean square deviation) values from 1 µs MD simulations, color-coded to match the configurations in (**A**). (**C**) Schematics illustrate the reaction coordinate in umbrella sampling, highlighting HS dissociation from H3 with a green force curve. (**D**) Histogram shows the binding free energies for HS-H3 interactions in Motifs 1, 2, 3, and the helical domain. (**E**) A free energy landscape map from a 1000 ns REMD (replica exchange molecular dynamics) simulation shows HS binding configurations to the helical and Mg(II) regions. (**F**) Panel provides salt bridge formation statistics between HS and H3's basic amino acids, with a bar chart of average formations. (**G**) A surface plot shows the frequency of salt bridge formations within H3, with areas of frequent formations in blue. (**H**) Detailed views of HS-H3 interactions, with the left image showing salt bridges and the right image displaying electrostatic interactions with Mg(II). This panel illustrates the impact on HS binding stability to H3 following the removal of Mg(II) during the simulation. (**J**) The effects of mutating all basic amino acids in the

*Figure 2 continued on next page*

*Figure 2 continued*

helical domain on the binding stability of HS are shown. (**K**) The 'palm-binding' model is depicted where HS is secured by the helical 'fingers' and interacts with the Mg(II)-bound 'palm'.

The online version of this article includes the following video, source data, and figure supplement(s) for figure 2:

**Source data 1.** RMSD (root mean square deviation) data used for *Figure 2B*.

**Source data 2.** Force-extension data used for *Figure 2C*.

**Source data 3.** Binding energy data used for *Figure 2D*.

**Source data 4.** Salt bridge data used for *Figure 2F*.

**Source data 5.** RMSD (root mean square deviation) data used for *Figure 2I*.

**Source data 6.** RMSD (root mean square deviation) data used for *Figure 2J*.

**Figure supplement 1.** Structural formula of heparan sulfate (HS).

**Figure supplement 2.** Docking results of H3 with heparan sulfate (HS).

**Figure supplement 3.** Umbrella sampling calculation of the binding free energy of HS-H3.

**Figure supplement 4.** Evolution of salt bridge formation in heparan sulfate (HS) during molecular dynamics (MD) simulations.

**Figure supplement 5.** Structure prediction of H3(uncharged) by AlphaFold2 (AF2).

**Figure 2—video 1.** Molecular dynamics (MD) simulation trajectories of heparan sulfate (HS) binding to H3 domain 1 and the helical domain.
https://elifesciences.org/articles/100545#fig2video1

**Figure 2—video 2.** Steered molecular dynamics (SMD) simulation trajectories of heparan sulfate (HS) dissociating from H3 in umbrella sampling.
https://elifesciences.org/articles/100545#fig2video2

**Figure 2—video 3.** Trajectories of H3 and heparan sulfate (HS) interactions in replica exchange molecular dynamics simulations (REMD).
https://elifesciences.org/articles/100545#fig2video3

**Figure 2—video 4.** Molecular dynamics (MD) simulation trajectories of the H3-H3 complex after removal of the Mg (II) from H3.
https://elifesciences.org/articles/100545#fig2video4

**Figure 2—video 5.** Molecular dynamics (MD) simulation trajectories of the H3-H3 complex after mutating the positively charged amino acids on the helical domain of H3 to serine.
https://elifesciences.org/articles/100545#fig2video5

number of salt bridges each amino acid formed during the simulations, further illustrating the intensity and frequency of these interactions. Our findings demonstrated a high affinity of HS for the helical domain of H3, with basic amino acids in this region forming more salt bridges compared to other parts of the protein. The surface hotspot map of H3 (*Figure 2G*) visually highlighted these interactions, with varying shades of blue representing the frequency of salt bridge formation, emphasizing their distribution and intensity. In the helical domain, a focus view showed that the 1st, 7th, 9th, and 18th sugar units of HS formed salt bridges with R248, R242, R267, and R259 of H3, respectively (*Figure 2H*, left), while the sulfate group on the 2nd sugar side chain of HS bound to Mg(II) (*Figure 2H*, right).

To further prove these findings, we performed two control experiments. First, we removed the Mg(II) ion during MD simulation. According to the results of umbrella sampling, we extended the simulation of the most stable H3-HS binding conformation to 1000 ns, then removed the Mg(II) ion and continued the simulations. The resulting increase in RMSD fluctuations confirmed the stabilizing role of Mg(II) in the binding process (*Figure 2I*, *Figure 2—video 4*). Second, we mutated all seven positively charged amino acids in the helical domain to serine, and name it H3 (uncharged). While AF2 predicted that the helical structure would remain intact (*Figure 2—figure supplement 5*), a significant alteration in HS binding dynamics was observed, supporting the critical role of these charged residues in stabilizing the interaction (*Figure 2J*, *Figure 2—video 5*).

Our interaction model revealed that the α-helical domain of H3 functions like a thumb, guiding HS into a stable binding position, while the Mg(II) region acts like a palm, securing HS in place. This dynamic interaction allows HS to transition from an initial surface-level binding to being deeply anchored within the protein cavity, illustrating a complex yet well-organized binding process (*Figure 2K*).

Bioinformatic analysis of the H3 protein across the Poxviridae family highlights its evolutionary conservation and its significance in HS binding. Given the crucial role of electrostatic interactions in the H3-HS binding process, we analyzed the charge distribution within the helical domain. To date, H3 proteins have been discovered in 66 of 118 known Poxviridae species. Using NCBI's BLAST, we

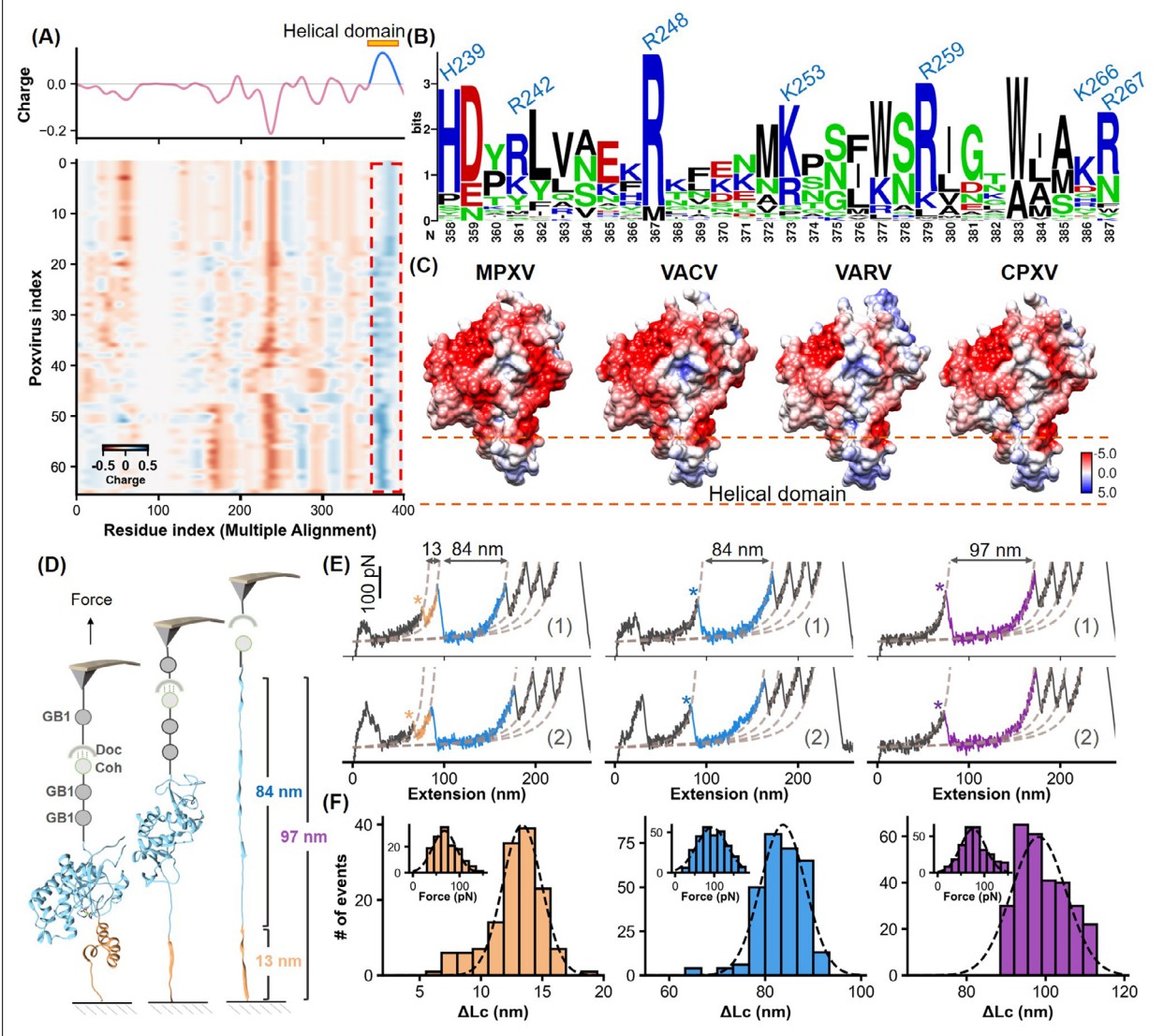

**Figure 3.** Charge characteristics and structural analysis of H3 protein. (**A**) The heatmap shows the amino acid charge distribution in the H3 protein across 66 Poxviridae viruses, following multiple sequence alignment. Blue indicates areas with more positive charges, and red indicates more negative charges. The accompanying curve shows the average charge of all amino acids in Poxviridae H3. (**B**) Logo plot of the amino acid sequence of the helical region, highlighting the conservation of basic amino acids at specific positions. (**C**) The surface charge analysis of H3 from the monkeypox virus (MPXV), vaccinia virus (VACV), variola virus (VARV), and cowpox viruses (CPXV) with the helical domain showing a significantly positive charge. (**D**) Schematic of the single-molecule force spectroscopy unfolding experiment on H3, illustrating the unfolding process of the helical domain (yellow) followed by the main body (blue). (**E**) Representative curves of H3 unfolding, color-coded to show the helical region (yellow), main body (blue), and full-length (purple) unfolding. (**F**) Histograms depict the force spectroscopy signals for helical domain (n=139), main body (n=294), and full-length unfolding (n=289), with ΔLc statistics provided. The inset shows a Gaussian fit of unfolding forces.

The online version of this article includes the following source data and figure supplement(s) for figure 3:

**Source data 1.** Evolutionary analysis data used for *Figure 3A*.

**Source data 2.** Evolutionary analysis data used for *Figure 3A*.

**Source data 3.** Unfolding ΔLc and force data used for *Figure 3F*.

**Figure supplement 1.** AlphaFold2 (AF2) structural prediction of 66 Poxviridae virus H3 proteins.

**Figure supplement 2.** Analysis results of full sequence analysis for 66 Poxviridae virus H3 proteins.

**Figure supplement 3.** Protein immobilization in atomic force microscopy-based single-molecule force spectroscopy (AFM-SMFS) experiments.

retrieved sequences of these 66 H3 proteins and performed multiple sequence alignment (*Altschul et al., 1997*). A heatmap detailing the side-chain charge distribution at a physiological pH of 6.5—mimicking the microenvironment of H3-HS interaction—revealed a significant concentration of positive charges at the helical domain (*Figure 3A*). Furthermore, structural predictions using AF2 showed that all these 66 sequences generally share a similar architecture to MPXV H3, particularly in the presence of the α-helical domain (*Figure 3—figure supplement 1*).

Notably, this domain consistently exhibited an accumulation of basic amino acids such as lysine, arginine, and histidine, which are essential for binding the negatively charged HS. These amino acids, including residues identified as 358, 361, 367, 373, 379, 386, and 387 in Poxviridae H3 (global position in alignment), corresponding to residues H239, R242, R248, K253, R259, K266, R267 in MPXV H3, show high conservation (*Figure 3B*, *Figure 3—figure supplement 2*). Surface charge analysis of H3 proteins from four *orthopoxviruses*, including MPXV, VACV, VARV, and CPXV, confirmed the predominance of strong positive charges in their α-helical domains (*Figure 3C*), crucial for effective binding to HS. This consistent feature across different viruses underscores the evolutionary importance of the helical domain, suggesting a universal mechanism in orthopoxviral adhesion that could be targeted in antiviral strategy.

## Experimental confirmation of the structure and function of α-helical domain

To validate the computational predictions, we first conducted experiments using atomic force microscopy-based single-molecule force spectroscopy (AFM-SMFS). This technique, widely used for studying protein (un)folding, allowed us to directly assess the structure and stability of the α-helical domain within the H3 protein (*Yu et al., 2017*; *Dietz and Rief, 2004*; *Goktas et al., 2018*). We engineered the H3 construct with two GB1 domains, each 18 nm in length upon unfolding, and one Cohesin module for precise measurement (*Figure 3D*; *Cao et al., 2006*; *Stahl et al., 2012*; *Zheng et al., 2023*). These proteins are immobilized on the surface by click chemistry and enzymatic ligations (*Figure 3—figure supplement 3*; *Shi et al., 2022*; *Deng et al., 2019*). During the experiment, the coated AFM tip with GB1 and Dockerin initiated a Coh-Doc interaction, producing a characteristic sawtooth-like force-extension curve as it was stretched (*Figure 3D and E*). Notably, two peaks corresponding to the unfolding of the α-helical domain were observed. The first peak showed a contour length increment (ΔLc) of 13 nm, closely matching the theoretical unfolding length for the 42 amino acid-long α-helical domain of H3 (residues 240–282, 42aa*0.36 nm/aa-2.6 nm) (*Carrion-Vazquez et al., 1999*). The measured unfolding force was 67.4±2.1 pN (mean ± SEM, n=139), while the second peak indicated the unfolding of the remaining structure of H3, with a ΔLc of 84 nm (*Figure 3F*). The total unfolding, represented by a cumulative ΔLc of 97 nm, confirmed the structured and stable nature of the α-helical domain within H3.

We further explored the functional role of the α-helical domain in mediating H3's binding to HS using AFM on live cells (*Tian et al., 2021*; *Alsteens et al., 2017*). The H3 protein was linked to the AFM tip and approached cultured Chinese hamster ovary K1 (CHO-K1) cells, which express HS, mounted on a Petri dish. Inverted microscopy was used to precisely control the positioning of the AFM tip. Upon contact, the interaction between H3 and HS was initiated, and the binding event was monitored by retracting the tip to generate a force-extension curve. A distinct peak, corresponding to H3-HS interaction, was observed with a dissociation force of 33.7±0.2 pN and a binding probability of 43% (*Figure 4A and B*). Repeated trials across different cell-surface areas produced a detailed force map, quantifying the distribution of HS unbinding forces and further confirming the critical role of the helical domain in viral adhesion (*Figure 4C and D*).

Moreover, we performed mutagenesis studies on the α-helical domain by replacing all positively charged residues with serine. AFM experiments on this variant H3(uncharged) showed a detectable force peak with a lower unbinding force of 28.8±0.2 pN, compared to 33.7±0.3 pN for the wild-type—representing a 14% reduction (ΔForce>2*SEM, 95% CI)—and a significant decreased HS binding probability from 43% to 25% (*Figure 4C and D*). These results suggest that the basic amino acids in the helical domain are crucial for HS binding. To confirm the specificity of the interaction, CHO-K1 cells were treated with heparinase II, an enzyme that degrades HS. This treatment further reduced the dissociation force to 23.0 pN and the binding probability to 12.6%, confirming that the observed interactions were specific to the HS-H3 complex.

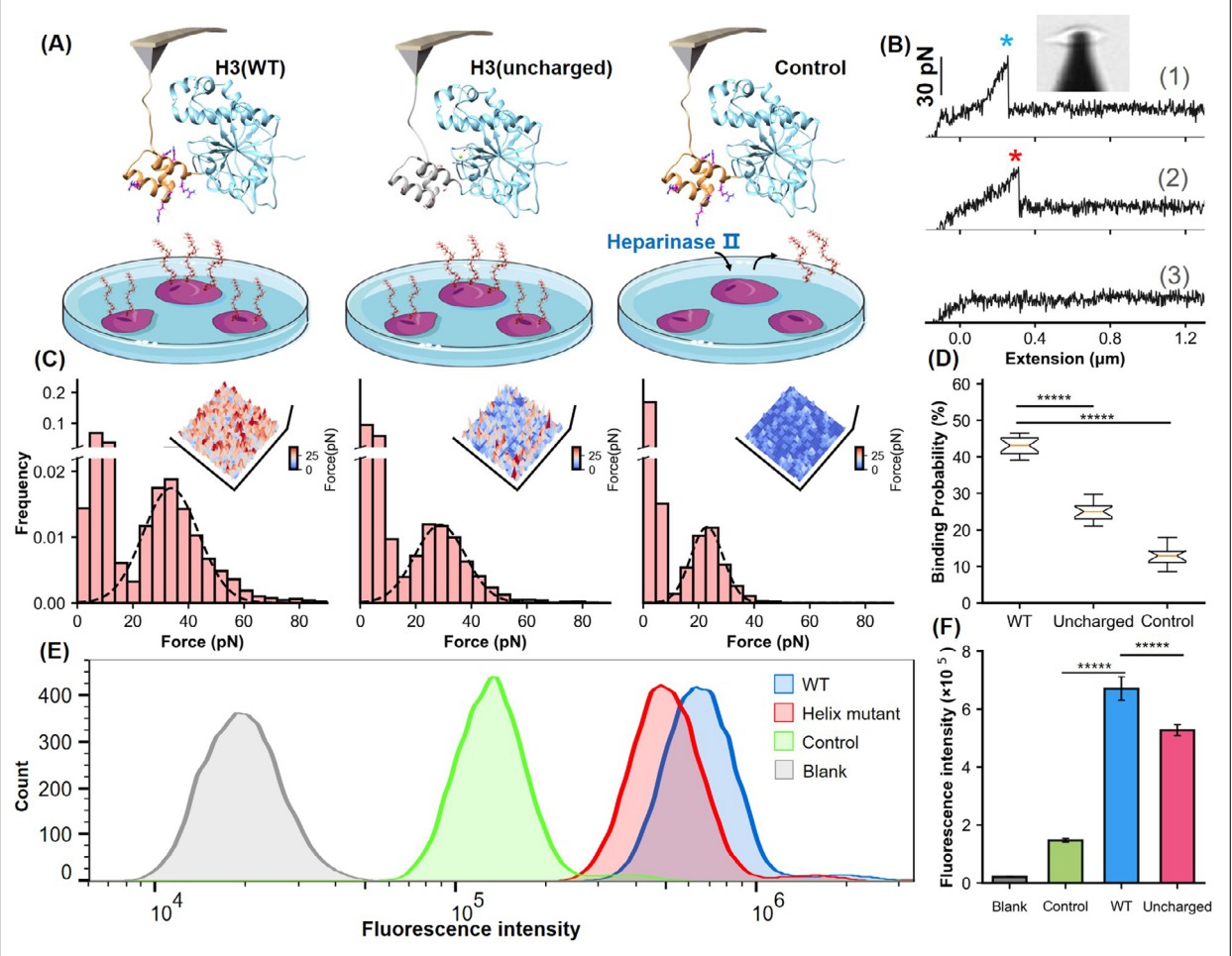

**Figure 4.** Analysis of the helical domain's interaction with heparan sulfate (HS) at cellular level. (**A**) Schematic of the cell force spectroscopy experiment setup shows three scenarios: wild-type H3 on an atomic force microscopy (AFM) tip interacting with HS on Chinese hamster ovary K1 (CHO-K1) cells, mutation of all basic amino acids in the H3 helical region to serine, and cells treated with HS hydrolase to remove surface HS before testing. (**B**) Force-extension curves depict interactions for the wild-type (WT), mutant, and control groups, marked with blue and red asterisks for dissociation events. An inset shows the optical microscope positioning the AFM probe. (**C**) Histograms of dissociation signals comparing the WT, mutant, and control groups, with an inset detailing the surface distribution of dissociation forces. (**D**) Statistical graph showing binding probabilities for different groups (WT n=18, Uncharged n=17, Control n=12), highlighting significant differences determined by Student's t-tests (p<1e-5). (**E**) Flow cytometry (FCM) results illustrate interactions of WT and uncharged H3 fused with eGFP with CHO-K1 cells, alongside GFP control (green) and cell-only control (blank, gray). (**F**) Statistical analysis of FCM data (n=10), showing significant differences between groups as determined by Student's *t*-test. *****, *P*<1e-5. Error bars indicate SD.

The online version of this article includes the following source data for figure 4:

**Source data 1.** Unbinding force data used for *Figure 4C*.

**Source data 2.** Atomic force microscopy-based single-molecule force spectroscopy (AFM-SMFS) unbinding probability data used for *Figure 4D*.

**Source data 3.** ECM binding data used for *Figure 4F*.

Further validation was carried out using flow cytometry (FCM). We fused eGFP, a fluorescent reporter protein, to the C-terminus of both the H3(WT) and the H3(uncharged). Following incubation of these fusion proteins with cells, FCM analysis showed a significant reduction in mean fluorescence intensity in cells treated with H3(uncharged) compared to H3(WT) (*Figure 4E*). Statistical analysis of multiple experiments highlighted a significant difference between the two (*Figure 4F*). These findings, along with the results from MD simulations and AFM force spectroscopy, underscore the importance of the positively charged amino acids in the α-helical domain of H3 for efficient HS binding.

## The helical domain serves as a target for H3 infection inhibition

Targeting the α-helical domain of H3 as H3 binding sites, we de novo designed a series of inhibitors. Using deep-learning model RFdiffusion, we designed a series of inhibitors for the H3-HS interaction and named it AI-PoxBlock (*Figure 5A*). Sequence recovery for each series was performed on 1000 backbones using ProteinMPNN, generating 10 sequences per backbone. These designs were further validated through complex structure predictions using AF2-multimer, and candidates were selected based on PAE (predicted aligned error) values below 7 and iPTM (interface predicted template modeling) scores above 0.9. The predicted structures of these candidates also exhibited RMSD values of less than 2 nm when compared to the RFdiffusion-generated structures (*Figure 5—figure supplement 1*). To assess the binding capabilities of these inhibitors, we conducted 500 ns MD simulations, analyzing RMSD trajectories. Five inhibitors—AI-PoxBlock302, 602, 614, 723, and 761—demonstrated stable RMSD values under 0.4 nm, confirming their potential for stable binding to the helical domain of H3 (*Figure 5—figure supplements 2 and 3*, *Figure 5—video 1*).

Although AI-PoxBlock302 could not be successfully expressed, the remaining four inhibitors underwent further testing in FCM to evaluate their efficacy in inhibiting H3 binding to CHO-K1 cell surfaces. AI-PoxBlock723 stood out, significantly reducing H3 binding at a concentration of 10 μM, while the other inhibitors displayed no tendency to inhibit H3 binding to HS on cell surfaces (*Figure 5B*, *Figure 5—figure supplement 4*). Further analysis using biolayer interferometry (BLI) confirmed that AI-PoxBlock723 binds to H3 with an equilibrium dissociation constant ($K_D$) of 9 μM (*Figure 5C*). To ensure specificity, we also assessed the interaction between AI-PoxBlock723 and a truncated version of H3 (residues 1–239), lacking the α-helical domain. BLI results indicated a marked reduction in binding, reinforcing the specificity of AI-PoxBlock723 for the helical domain (*Figure 5—figure supplement 5*). Circular dichroism (CD) spectroscopy also confirmed that AI-PoxBlock723 predominantly consists of α-helices, consistent with its design specifications (*Figure 5—figure supplement 6*).

To evaluate the antiviral activity of AI-PoxBlocks, the inhibitors were serially diluted and incubated with the clade II MPXV isolate SZTH42 before infecting Vero E6 cells. After overnight culture, virus infection foci were determined as previously described (*Cheng et al., 2024b*). Notably, AI-PoxBlock723 showed an antiviral effect with the half maximal inhibitory concentration ($IC_{50}$) of 88.6 μM, whereas other AI-PoxBlocks exhibited no inhibitory efficiency (*Figure 5D*). Given the high conservation of the H3 helical domain across the four *orthopoxviruses* pathogenic to humans (*Figure 5—figure supplement 7*), we also assessed AI-PoxBlock activity against VACV infection. Consistently, AI-PoxBlock723 is also effective for inhibiting VACA infection in vitro with a similar IC50 of 86.7 μM (*Figure 5—figure supplement 8*), suggesting the helical domain as a universal antiviral target for *orthopoxvirus*.

To evaluate the efficacy of AI-PoxBlock723 in vivo, groups of BALB/c mice were challenged with $4×10^5$ FFU (focus-forming unit) MPXV/SZTH42 as we previously reported (*Cheng et al., 2024a*). Mice were administered intraperitoneally with single dose of AI-PoxBlock723 (10 mg/kg) or phosphate buffer solution (PBS) immediately after MPXV challenge, and were sacrificed 4 days post-infection for the determination of pulmonary MPXV titers. The MPXV infectious particles and genome copies in lungs of the AI-PoxBlock723-treated mice were significantly lower than that in the control mice (*Figure 5E, F*, *Figure 5—figure supplement 9*).

These results demonstrate the efficacy of the designed inhibitors targeting the H3 helical domain and validate the crucial role of this domain in viral adhesion, providing a promising foundation for the development of novel antiviral agents.

## Discussion

Our study provides new mechanistic insights into MPXV infection by identifying the α-helical domain of the H3 protein as a critical mediator of HS binding. This domain, previously uncharacterized, plays a pivotal role in viral adhesion to host cells, a key step in MPXV pathogenesis. Using a combination of AI-based tools and MD simulations, we successfully identified the specific interaction sites between H3 and HS, offering a novel target for antiviral drug design (*Monzon et al., 2024*).

Previous studies have highlighted the importance of viral adhesion proteins in *orthopoxvirus* infection, but the specific mechanisms underlying the H3-HS interaction remained unresolved. Our findings provide the first detailed structural insight into this interaction, confirming the importance of the electrostatic nature of H3-HS binding. These results align with prior reports indicating the role of HS

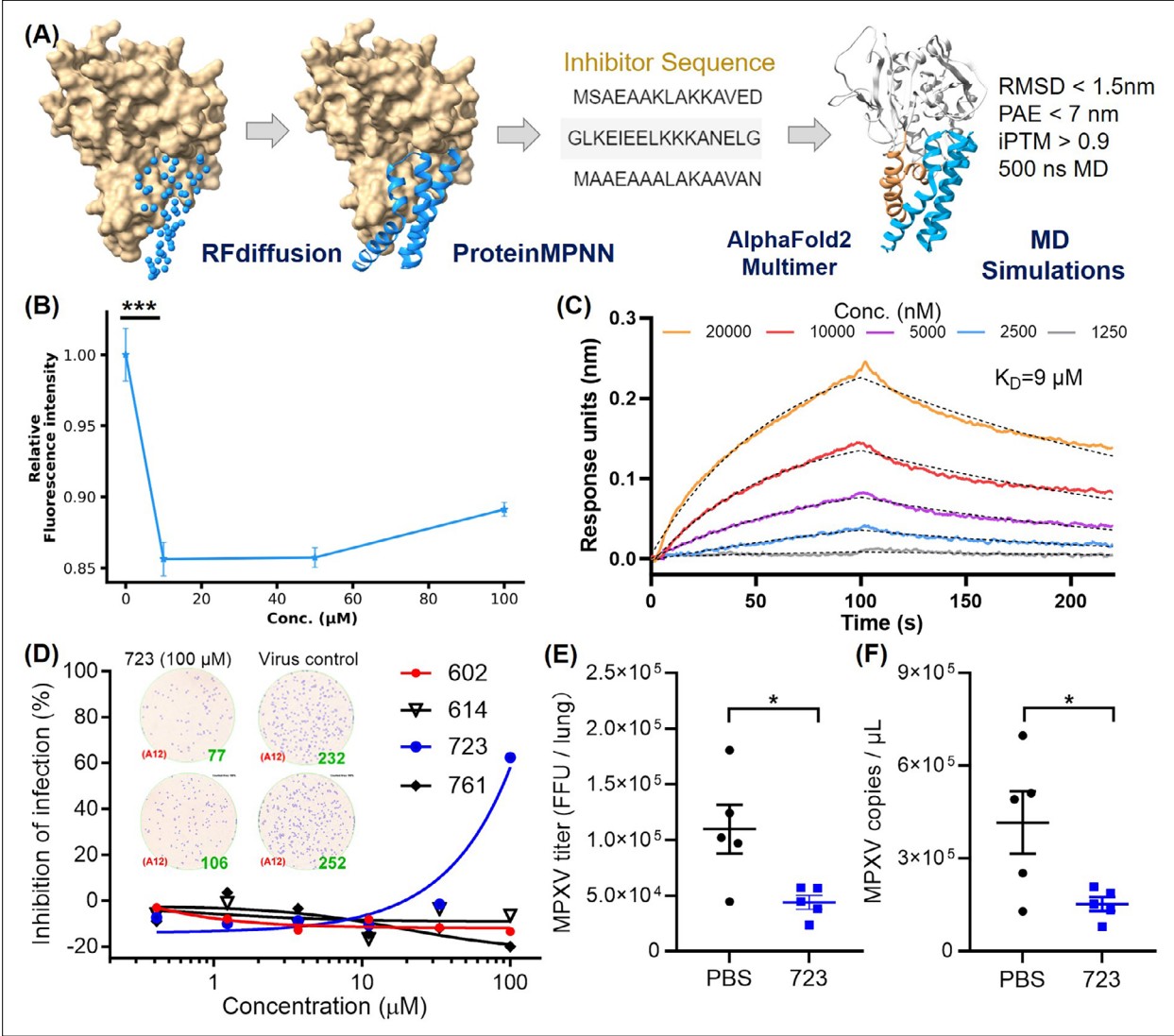

**Figure 5.** Inhibitor design targeting the H3-HS interaction and efficacy of AI-PoxBlock723. (**A**) Diagrams depict protein inhibitors designed to target the H3 helical region, created using RFdiffusion. Sequences capable of folding into the target scaffold structures were generated using ProteinMPNN, and were validated through AlphaFold2 (AF2), followed by 500 ns molecular dynamics (MD) simulations for structural stability and interaction scoring. (**B**) Flow cytometry (FCM) analysis demonstrates the inhibitory effect of AI-Poxblock723 at various concentrations (0, 10, 50, and 100 µM, x-axis, n=5). The control group consisted of 2 µM H3-eGFP without the inhibitor, with the relative fluorescence intensity normalized to 1. (**C**) Biolayer interferometry (BLI) confirms direct interaction between AI-Poxblock723 and the H3 helical domain. (**D**) Graphs display the inhibitory effect of the indicated AI-Poxblocks on monkeypox virus (MPXV) infection of Vero E6 cells, with quantitative analysis of virus-infected foci provided on the right. (**E–F**) BALB/c mice were infected intranasally with $4 \times 10^5$ FFU (focus-forming unit) MPXV and treated with single dose of AI-PoxBlock723 (10 mg/kg) or phosphate buffer solution (PBS) immediately after challenge (n=5). Infectious MPXV particles and MPXV viral loads in mice lungs at 4 days post-infection (dpi) were determined by focus-forming assay (**E**) and quantitative polymerase chain reaction (qPCR) (**F**). Error bars indicate SEM and statistical significance was evaluated using Student's $t$-test (B, E). *$P < 0.05$, ***$P < 0.001$.

The online version of this article includes the following video, source data, and figure supplement(s) for figure 5:

**Source data 1.** Inhibitory effect data used for *Figure 5B*.

**Source data 2.** Biolayer interferometry (BLI) data used for *Figure 5C*.

**Source data 3.** Inhibitory effect data used for *Figure 5D*.

**Source data 4.** Inhibitory effect data used for *Figure 5E*.

**Source data 5.** Inhibitory effect data used for *Figure 5F*.

**Figure supplement 1.** Artificial design of H3 inhibitors.

**Figure supplement 2.** AlphaFold2 (AF2)-predicted structures of the H3 and inhibitor complexes.

*Figure 5 continued on next page*

in viral entry, but advance the field by identifying specific residues within the α-helical domain that are crucial for binding.

One of the strengths of this study lies in the application of advanced AI-based tools, such as AF2 and RFdiffusion, alongside MD simulations. These cutting-edge computational approaches allowed us to overcome the limitations imposed by the dynamic and flexible nature of HS. In addition, HS serves as a cellular ligand for various viruses (*Clausen, 2020*; *Shriver et al., 2012*; *Koehler et al., 2020*), underscoring the importance of studying its interactions with viral adhesion proteins. By predicting the structure-function relationship of the H3-HS interaction, we were able to pinpoint key binding residues and mechanisms that were previously inaccessible. This demonstrates the growing potential of AI in structural biology, particularly for the complex protein-glycan interactions (*Shriver et al., 2012*; *Shih et al., 2009*; *Marszalek et al., 2003*).

The identification of the α-helical domain in H3 as a key player in HS binding offers a promising new target for antiviral drug development (*Gulati et al., 2022*; *Zhang et al., 2009*; *Podduturi et al., 2024*; *Bernardi et al., 2019*; *Zhao and Woodside, 2021*; *Serdiuk et al., 2016*; *Sen Gupta et al., 2023*; *Petrosyan et al., 2021*; *Chen et al., 2023*). Given the conservation of this domain across *orthopoxviruses*, including VARV and VACV, targeting this region could lead to the development of broad-spectrum antivirals effective against multiple *orthopoxviruses* (*Wang et al., 2023*). Our study highlights the power of AI-driven drug design, as demonstrated by the promising results of AI-PoxBlock723, which showed efficacy in inhibiting viral infection. Further optimization and testing of these inhibitors in clinical models will be critical steps toward developing effective antiviral therapies. For example, conjugate the peptides with carrier molecules, such as liposomes, nanoparticles, or dendrimers, which can protect the peptides from immune detection and improve their delivery to target cells.

Looking ahead, the methodologies and findings from this study are likely to catalyze further investigations into viral adhesion mechanisms, not only in *orthopoxviruses* but also across other viral families that utilize similar adhesion strategies. Although challenges remain, we hope our discovery will attract structural biologists to resolve this domain's structure in H3, and even the structure of H3-HS complex, to further enhance drug design efficiency for this target. Although this domain itself cannot express, we chemically synthesized it (residues 240–279). CD measurement on this fragment indeed showed an α-helical structure (*Figure 5—figure supplement 10*). As AI-driven approaches continue to evolve, they are poised to play a transformative role in advancing our understanding of viral pathogenesis and revolutionizing the future of drug discovery.

## Materials and methods
### Protein expression and purification

All genes were ordered from General Biosystems. H3(1–282) was constructed into the pCold-TF-tev vector using *BamH*I-*Bgl*II-*Kpn*I three-restriction enzyme system and a Strep-Tag II was added to the C-terminus for protein purification. eGFP was ligated to the C-terminus of H3(1–282) by restriction enzyme system. Mutations in the basic amino acids of the helical domain were introduced by

site-directed PCR mutagenesis. The pCold-TF-tev-POI-strep tag construct was transformed into BL21(DE3), and single colonies were picked and cultured overnight at 37°C in LB medium with ampicillin. The expanded culture was then transferred to 800 mL of ampicillin LB and continued to be incubated at 37°C with shaking for 2 hr. When the $OD_{600}$ of the culture reached 0.6–0.8, it was cooled to 15°C in an ice-water bath. Then, IPTG was added to a final concentration of 0.5 mM and incubated overnight at 15°C on a shaker.

The next day, the cells were collected by centrifugation at 8000 rpm, the supernatant was discarded, and the cell pellet was resuspended in buffer (50 mM Tris-HCl, 150 mM NaCl, pH 7.4). The cell suspension was then lysed with a homogenizer for 3 min, and the lysate was centrifuged at 18,000 rpm to separate the precipitate. TEV protease was added to the supernatant, along with EDTA and DTT to a final concentration of 1 mM each, and incubated at 30°C for 3 hr to remove the TF tag. The digested product was subjected to another high-speed centrifugation at 18,000 rpm, and the supernatant was collected. The supernatant was then purified using STarm Sreptactin. The purity of the obtained protein was verified by SDS-PAGE, and the final purity of the H3 protein exceeded 60%.

The proteins used for AFM force spectroscopy were sequentially constructed into the pQE80L vector using the same restriction enzyme system, resulting in pQE80L-coh-(GB1)$_2$-H3(1–282)-strep tag II-NAL. The protein expression method was similar to the one mentioned above. The difference was that after adding IPTG to a final concentration of 0.5 mM, the culture was incubated overnight at 18°C. After cell lysis and centrifugal removal of the precipitate, the proteins were purified using STarm Sreptactin. The eluate was exchanged into AFM working buffer (50 mM Tris-HCl, 150 mM NaCl, pH 7.4) using an ultrafiltration tube and then used for AFM force spectroscopy experiments.

*Oa*AEP1(C247A) is cysteine 247 to alanine mutant of asparaginyl endoproteases 1 from *Oldenlandia affinis*, abbreviated as AEP here. ELP is the elastin-like polypeptides. Their expression and purification protocols can be found in references (*Shi et al., 2022*; *Deng et al., 2019*; *Liu et al., 2024*). The TEV protease was expressed in the pCold-His6-ProS2 vector (the original tev protease cleavage site in the vector was removed by PCR). The expression was induced with a final concentration of 0.1 mM IPTG and incubated overnight at 15°C in a constant temperature shaker. The His6-ProS2-TEV was purified using Ni-NTA affinity chromatography.

All H3 inhibitors designed via RFdiffusion were constructed into the pET30a vector, with C-terminal fusion of strep tag and His8 tag for protein purification and subsequent BLI immobilization. After transforming pET30-Inhibitor-strep tag II-His8 into BL21(DE3), the culture was grown at 37°C in kanamycin-resistant LB medium for 2 hr until the $OD_{600}$ reached 0.6–0.8, then IPTG was added to a final concentration of 0.5 mM and the culture was incubated overnight at 18°C. The subsequent purification steps were as previously described, using STarm Sreptactin. The purified protein was exchanged into PBS buffer via ultrafiltration. 4 mg chemically synthesized peptide H3 (240–279) was ordered from GenScript Corporation.

## Modeling and MD simulations of C-terminal insertion of the H3 into DPPC membrane

Using TMHMM2.0, we predicted the transmembrane region of H3 to span residues 283–306. We modeled this region using ESMFold, obtaining the helical structure of H3(283–306). Subsequently, by utilizing the previously obtained structures of H3(1–282) and H3(283–306), we performed multi-template modeling of the full-length H3 protein using MODELLER. Next, we employed CHARMM-GUI's (*O'Toole et al., 2023*) Membrane Builder to construct the structure of H3 inserted into a 10×10 nm$^2$ DPPC membrane, and simultaneously built a water box with a NaCl concentration of 0.15 M, generating input files (charmm36m force field and the TIP3P water model) suitable for MD simulations with GROMACS 2023.3.

After energy minimization to reduce the maximum force below 1000 kJ/mol/nm, an NVT equilibration was conducted using the V-rescale thermostat method at 310 K. This was followed by a 500 ns NpT production simulation at 310 K using the V-rescale thermostat and the C-rescale barostat methods at 1 bar. During the 500 ns MD simulation, the time-dependent changes in the minimum distance between the helical region H3(240–273) and the DPPC membrane were calculated from the simulation trajectory using *gmx mindist*.

## Molecular docking of H3 with HS

An HS oligosaccharide with the structure -[IdoA2S-GlcNS6S-IdoA-GlcNS(3,6S)]$_5$- was modeled using the CHARMM-GUI online website. The structure of H3(1–282) predicted by AF2 was refined through 500 ns of MD simulation, and the most frequently appearing conformation was obtained through structural clustering. Specifically, by calculating the RMSD values of H3 in the MD trajectory, and then clustering using the built-in gmx cluster program, the gromos clustering method was adopted with a cutoff set to 0.2 nm. The top three conformations obtained were used for the subsequent docking process.

AutoDock Vina was called via the Chimera graphical interface to generate docking configuration files for HS and H3, with the center of each docking box set at the center of the GAGs binding motif, and the box size set to 8 nm. The energy_range was set to 3 kcal/mol, specifying that the energy difference from the current best conformation should be less than 3 kcal/mol. The docking results were ranked according to the energy scores generated by AutoDock Vina, and the best four conformations were selected.

## MD simulations of the H3-HS complex

The docked H3-HS complex was uploaded to the CHARMM-GUI website for simulation system construction, utilizing the charmm36m force field and the TIP3P water model. A water box, 1 nm larger than the molecular boundaries of the complex, was constructed, and 0.15 M NaCl was used for charge balancing. The final output included structure and force field files suitable for GROMACS 2023.3 MD simulations. After energy minimization to reduce the maximum force below 1000 kJ/mol/nm, an NVT equilibration was conducted using the V-rescale thermostat method at 310 K for 1 ns. This was followed by a 1000 ns NpT production simulation at 310 K using the V-rescale thermostat and the C-rescale barostat methods at 1 bar. Three independent repetitions were performed for the production simulation.

The RMSD values of HS were calculated using the built-in gmx rms program in GROMACS, maintaining the structural overlap of H3. To analyze salt bridges between HS and HS in this simulation, a Python script was used to calculate the distance between the negatively charged oxygen atoms on the side chains of HS and the nitrogen atoms on the side chains of basic amino acids of H3 throughout the trajectory. A distance less than the set cutoff of 0.35 nm was considered indicative of salt bridge formation.

## Binding free energy of H3 with HS using umbrella sampling in MD simulations

To calculate the stability of various binding conformations of H3-HS, the binding free energy was determined using the umbrella sampling method. The force field and solvent settings of the simulation system were consistent with the previously mentioned MD simulations, employing the charmm36m force field and TIP3 water model, and maintaining a simulation temperature of 310 K. These simulations were conducted using GROMACS 2023.3. Specifically, for umbrella sampling, steered molecular dynamics simulations were first carried out. During this process, H3 was held fixed (using a restraining potential), and a harmonic potential was applied to the HS molecule. This potential facilitated the constant velocity stretching of HS (1 nm/ns) along the z-axis, moving it 5 nm until complete separation from H3 was achieved. The size of the simulation box was designed to ensure that neither molecule crossed the boundaries during stretching. The stretching process generated a series of reaction coordinates, defined by the distance between the centroids of HS and H3. Biasing potentials were applied to these coordinates for 10 ns MD simulations. The outputs of these simulations, in the form of potential energy distributions, were then analyzed using the weighted histogram analysis method. This analysis produced a PMF profile, describing the free energy landscape as a function of the reaction coordinate, thus allowing for the calculation of the binding free energy of H3-HS.

## REMD simulations

To further investigate the conformational changes of HS and the helical domain, we conducted REMD simulations. The simulation system was set up using CHARMM-GUI, employing the same force field and water model as the aforementioned MD simulations, and 0.15 M NaCl was used for charge balancing. The temperature range was set from 310 K to 387 K, generating a total of 64 replicas with

a swap probability of 20%. Each simulation had a duration of 1 μs, resulting in a cumulative simulation time of 64 μs for all replicas.

Simulations were executed using the MPI version of GROMACS 2023.3. Upon completion of the simulations, the replica at 310 K was selected for further analysis. Using GROMACS built-in tools, 'distance' and 'rms', we calculated the nearest distance between HS and Mg(II), and the RMSD of HS throughout the simulation trajectory, respectively. Subsequently, the gmx sham program was used to generate a free energy landscape from these two sets of data. This landscape effectively illustrates the dynamic binding changes between HS and H3, as well as the interaction between HS and Mg(II).

## Poxviridae virus H3 sequence analysis

A total of 66 available H3 sequences from Poxviridae viruses were obtained through a sequence search of the MPXV H3 sequence using NCBI's BLAST. After removing duplicate sequence data, multiple sequence alignment was performed using MEGA11, with the ClustalW method employed for sequence alignment. The alignment results were uploaded to the WebLogo website for logo image generation. Acidic amino acids, basic amino acids, polar amino acids, and non-polar amino acids were represented in red, blue, green, and black, respectively. The font size in the logo corresponds to the probability of occurrence of the amino acid at that position, with larger fonts indicating greater conservation of the amino acid. The charge analysis of the amino acid sequences was conducted using the ProteinAnalysis function of the Biopython package. Surface charge maps of the H3 protein were generated using Chimera, and the charge distribution was calculated using APBS.

## AFM-SMFS protein unfolding experiment

The AFM cantilever/tip made of silicon nitride (MLCT-BIO-DC, Bruker Corp.) was used. The detailed protocol for AFM tip functionalization and protein immobilization on the glass coverslip can be found in the literature (*Liu et al., 2024*). In short, the tip and glass coverslip were coated with the amino group by amino-silanization. $N_3$ is functionalized on the surface from the reaction between $ImSO_2N_3 \cdot HCl$ and $-NH_2$. Then, a heterobifunctional DBCO-PEGn-Mal can be reacted and adds the Mal group. Next, the peptide $GL\text{-}ELP_{20}\text{-}C$ or $C\text{-}ELP_{20}\text{-}NGL$ was reacted to the maleimide via the cysteine, respectively. The long $ELP_{20}$ serves as a spacer to avoid non-specific interaction between the tip and the surface as well as a signature for the single-molecule event. Finally, target protein H3 with C-terminal NGL sequence or GB1-Doc with N-terminal GL sequence can be site-specifically linked to the coverslip or tip by AEP, respectively.

Atomic force microscope (Nanowizard4, JPK) was used to acquire the force-extension curve. The D tip of the MLCT-Bio-DC cantilever was used. Its accurate spring constant was determined by a thermally induced fluctuation method. Typically, the tip contacted the protein-immobilized surface for 100 ms under an indentation force of 350 pN to ensure a site-specific interaction. Then, moving the tip up vertically at a constant velocity (1 μm/s), the polyprotein unfolded. Then, the tip moved to another place to repeat this cycle several thousands of times. As a result, a force-extension curve was obtained, which was analyzed using JPK data process analysis software.

## AFM-SMFS unbinding experiment of H3-HS on CHO-K1

Force-extension-based AFM measurement on model surfaces was performed in PBS buffer at room temperature using functionalized D tip of MLCT-Bio-DC cantilever (Bruker, nominal spring constant of 0.030 N/m and actual spring constants calculated using thermal tune). AFM (Nanowizard4, JPK) operated in the force mapping (contact) mode was used. CHO-K1 cells were cultured in a 37°C $CO_2$ incubator for 24 hr prior to force spectroscopy experiments. The relative positioning of the AFM probe and the adherent cells was determined using an inverted microscope. Areas of 5×5 μm² were scanned, ramp size set to 350 nm, and set point force of 300 pN with a contact time of 200 ms, with a resolution of 32×32 pixels.

## Flow cytometry experiment

Protein binding to GAGs expressed on the surface of CHO-K1 cells was assessed using FCM, following the previously described methods. H3-eGFP, H3(uncharged)-eGFP, and eGFP proteins (5 μM) were incubated with 1 million cells (200 μL) at 4°C for 20 min. Post-incubation, cell analysis was performed using the CytoFLEX flow cytometer (Beckman) to collect fluorescence signals in the FITC channel. A

total of 15,000 events were collected using the CytExpert software (version 2.5; Beckman), with 10 sets of data collected in parallel. The data from the live-cell gating were analyzed using FlowJo software (version 10.6.2; Tree Star, San Carlos, CA, USA).

The FCM experiment to validate the inhibitory effects of inhibitors utilized the same equipment and methods as previously described. The final concentrations of H3-eGFP and the inhibitor were 2 μM and 10 μM, respectively, with a total volume of 600 μL. The control group contained only 2 μM H3-eGFP, with the volume made up with PBS. Data from five parallel experiments were collected, with each experiment gathering 15,000 events. Using FlowJo, the fluorescence intensity of the mean of FITC-area signal in the live-cell gate was analyzed, normalizing the relative fluorescence intensity of the control group to 1.

To further verify the concentration-dependent inhibitory effect of AI-Poxblock 723, we conducted a concentration-dependent FCM experiment. The final concentration of H3-eGFP remained at 2 μM, while 723 was tested at three final concentrations (100 μM, 50 μM, and 10 μM), with a total volume of 600 μL. The control group contained only 2 μM H3-eGFP without inhibitor. Data from five parallel experiments were collected, with 15,000 events recorded for each experiment. The mean fluorescence intensity in the FITC-area signal within the live-cell gate was analyzed using FlowJo software, normalizing the relative fluorescence intensity of the control group to 1.

## Design of inhibitors targeting the H3 helical domain

To design inhibitors targeting the helical domain of H3, hotspots information was first inputted into RFdiffusion. The selected residues for this purpose were 239, 242, 267, 266, 259 of H3, and the length of the inhibitor sequences was set to be 40–80 amino acids. This process generated 1000 backbone structures. After excluding single-stranded helical structures, the remaining 633 structures underwent sequence recovery using ProteinMPNN, producing 10 sequences per structure. All 6330 sequences were then subjected to structure prediction and scoring using AF2. Structures with an iPTM score greater than 0.9 and a PAE score less than 6 were selected, while sequences containing Cys were excluded to avoid the formation of disulfide bonds.

RMSD calculations were performed between the predicted structures and the RFdiffusion-designed structures, and sequences with an RMSD greater than 1.5 nm were excluded to ensure consistency between the predicted and designed structures. Additionally, solubility assessments were conducted for all sequences. Specifically, a solubility parameter was assigned to each of the 20 natural amino acids, and an average was calculated for the entire sequence. The sequences with the highest solubility were selected for expression and interaction testing.

## MD simulations of H3 and its inhibitors

The AF2-predicted structure of the H3-inhibitor complex was used for the MD simulations, utilizing the previously mentioned MD software and force field. The MD simulation was conducted at 310 K for 500 ns. Subsequently, the RMSD of the simulation trajectory was calculated using the built-in 'rms' program of GROMACS to analyze the stability of the inhibitor-H3 binding.

## CD experiments

For CD experiments, AI-Poxblock723 and H3(240–279) were diluted to 0.15 mg/mL and 0.1 mg/mL in 10 mM K-PO$_4$ (pH 7.4) buffer, respectively. Spectra were acquired on a Chirascan V100 (Applied Photophysics). The data acquisition wavelength is set to 190–260 nm. All reported measurements were acquired within the linear range of the instrument.

## BLI binding experiments

BLI experiments were conducted using Octet HIS1K Biosensors on the Octet system. The buffer used throughout the experiments was uniform PBS. The tips were first pre-equilibrated in PBS solution for 10 min, followed by a 60 s baseline, 60 s loading, 100 s baseline, 100 s association, and 120 s dissociation steps. Baseline measurements of unloaded tips were subtracted from their matched measurement of the loaded tip. The inhibitor concentration was determined using the BCA method before loading and prepared at a concentration of 0.1 mg/mL for the loading solution. The mass concentration of H3(1–282) and H3(1–239) was also determined using the BCA method, and their molar concentrations were calculated. The purity of H3 is above 60%. Various concentration samples were

then prepared using a serial dilution method. After baseline correction and curve smoothing, a global fit was performed to calculate the $K_D$.

## Inhibition test for viral infection

Vero E6 cells were seeded at $1.5\times10^4$ cells/well into 96-well plates and used the following day. AI-Pox-blocks threefold serial diluted in maintenance medium were mixed with an equal volume of diluted live VACV or MPXV and then incubated at 37°C for 1 hr. Medium from 96-well plates was aspirated, and the inhibitor-virus mixture was added (100 μL/well), then the Vero E6 cells were incubated at 37°C for about 16 hr. Then cells were fixed with 4% paraformaldehyde solution, permeabilized with Perm/Wash buffer (BD Biosciences) containing 0.1% Triton X-100, incubated with the HRP-conjugated anti-VACV polyclonal antibodies (Invitrogen) diluted in the Perm/Wash buffer at room temperature for 2 hrs.

The reactions were developed with KPL TrueBlue Peroxidase substrates (Seracare Life Sciences). The numbers of infected foci were calculated using an EliSpot reader (Cellular Technology Ltd). The 50% inhibitory concentration (IC50) was calculated using GraphPad Prism software using the log (inhibitor) vs. normalized response-variable slope (four parameters) model. Experiments with live MPXV were performed in a Biosafety Level 3 (BSL-3) facility following standard biosafety practices.

## Animal experiments

Female BALB/c mice, 6–8 weeks of age, were obtained from GemPharmatech Co., Ltd (Guangdong, China). After arrival, all mice were acclimated for 3 days and then randomly assigned to experimental groups. For challenge, mice were anesthetized by intraperitoneal injection of 1.25% tribromoethanol solution with a dosage of 20 μL/g. Intranasal infections were performed by introduction of 50 μL of virus into one nostril. Mock-infected control animals were similarly inoculated with an equivalent volume of diluent. After infection, mic e were given intraperitoneally with single dose of 10 mg/kg of AI-Poxblock723 or PBS. Animals were euthanized 4 days after infection for the determination of pulmonary viral titers.

## Animal ethics statement

Mice were housed in independent ventilation cages and utilized at five mice per treatment group. All animals were given food and water ad libitum throughout all experiments. All efforts were made to minimize animal suffering and to reduce the number of animals used. All animal procedures were approved by Shenzhen Third People's Hospital's Institutional Animal Care and Use Committee prior to the initiation of studies.

## Infectious virus titration and viral load determination

The lungs from BALB/c mice were homogenized in 1 mL PBS with 3 mm zirconium beads using a tissue homogenizer (Omni, Bead Ruptor 24 Elite). After two cycles of freeze-thaw, the lung tissue homogenates were centrifuged to remove tissue debris. Supernatants were transferred into fresh tubes. Infectious viral titers were determined using focus-forming assay. Briefly, the Vero E6 cells were seeded in 96-well plates and incubated overnight. Ten-fold serial dilutions of the MPXV were prepared in the maintenance medium. The Vero E6 cells were inoculated with 100-fold diluted supernatants (100 μL/well). After 1 hr of adsorption, the samples were removed and cells were washed once with PBS. Then 100 μL/well maintenance medium was added. After 18 hrs incubation, the medium was aspirated and the cells were fixed with 4% paraformaldehyde solution for 30 min, permeabilized with Perm/Wash buffer (BD Biosciences) containing 0.1% Triton X-100, incubated with the HRP-conjugated anti-VACV polyclonal antibodies (Invitrogen) at room temperature for 2 hrs. The reactions were developed with KPL TrueBlue Peroxidase substrates (Seracare Life Sciences). The number of virus foci was counted using an ELISpot reader (Cellular Technology Ltd.).

MPXV viral loads (genome DNA copies) were determined using quantitative polymerase chain reaction (qPCR). Briefly, MPXV DNA in tissue homogenate supernatants was extracted using a nucleic acid extraction instrument (DaAn Gene, Smart 32) combined with Daan's extraction kit according to the manufacturer's instructions. The MPXV DNA copies were determined using SYBR Green Premix Pro Taq HS qPCR Kit (Accurate Biotech) with the following primers: forward primer

5'-TTT ATT CAA CAT GTA CTG TAC CCA C-3', reverse primer 5'- TTT CTT GCA TGG ATT TTC GTA TTT C -3'. MPXV B6R plasmid (Sangon Biotech) was serially diluted and performed to generate the standard curve.

## Acknowledgements

We appreciate the inspiring comments from previous anonymous reviewers. We acknowledge the funding support from the National Natural Science Foundation of China (22222703, 22477058), the Fundamental Research Funds for the Central Universities (020514380335), the Natural Science Foundation of Jiangsu Province (BK20202004), Shenzhen Medical Research Funds (B2302052). The numerical calculations in this work have been done on the computing facilities in the High-Performance Computing Center (HPCC) of Nanjing University.

## Additional information

### Funding

| Funder | Grant reference number | Author |
|---|---|---|
| National Natural Science Foundation of China | 22222703 | Peng Zheng |
| Fundamental Research Funds for the Central Universities | 020514380335 | Peng Zheng |
| Natural Science Foundation of Jiangsu Province | BK20202004 | Peng Zheng |
| Shenzhen Medical Research Funds | B2302052 | Lin Cheng |
| National Natural Science Foundation of China | 22477058 | Peng Zheng |

The funders had no role in study design, data collection and interpretation, or the decision to submit the work for publication.

### Author contributions

Bin Zheng, Data curation, Formal analysis, Investigation, Methodology, Writing – original draft; Meimei Duan, Investigation, Methodology; Yifen Huang, Shangchen Wang, Jun Qiu, Zhuojian Lu, Lichao Liu, Guojin Tang, Investigation; Lin Cheng, Resources, Supervision, Funding acquisition, Project administration, Writing – review and editing; Peng Zheng, Conceptualization, Resources, Supervision, Funding acquisition, Validation, Writing – original draft, Project administration, Writing – review and editing

### Author ORCIDs

Lin Cheng http://orcid.org/0000-0001-8066-527X
Peng Zheng https://orcid.org/0000-0003-4792-6364

### Ethics

Animal ethics statement was described in the Methods section. This study was approved by the Experimental Animal Ethics Committee of the Shenzhen Third People's Hospital, China (approval number: 2023-019).

Reviewer #1 (Public Review): https://doi.org/10.7554/eLife.100545.3.sa1
Reviewer #2 (Public Review): https://doi.org/10.7554/eLife.100545.3.sa2
Reviewer #3 (Public Review): https://doi.org/10.7554/eLife.100545.3.sa3
Author response https://doi.org/10.7554/eLife.100545.3.sa4

## Additional files

**Supplementary files**
MDAR checklist

**Data availability**
All data are available in the main text or the supplementary materials.

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
