## [Editor Report · eLife Assessment]

This work presents **important** findings regarding the interaction of the monkeypox virus (MPXV) attachment H3 protein with the cellular receptor heparan sulfate and the use of this information to develop antivirals potentially effective against all orthopoxviruses. Using a combination of state-of-the art computational and wet experiments the authors present **convincing** evidence to sustain their claims. These results will interest those working on basic orthopoxviruses biology and antiviral development.

---

## [Referee Report · Reviewer #1 (Public Review)]

Summary:

The study aimed to better understand the role of the H3 protein of the Monkeypox virus (MPXV) in host cell adhesion, identifying a crucial α-helical domain for interaction with heparan sulfate (HS). Using a combination of advanced computational simulations and experimental validations, the authors discovered that this domain is essential for viral adhesion and potentially a new target for developing antiviral therapies.

Strengths:

The study's main strengths include the use of cutting-edge computational tools such as AlphaFold2 and molecular dynamics simulations, combined with robust experimental techniques like single-molecule force spectroscopy and flow cytometry. These methods provided a detailed and reliable view of the interactions between the H3 protein and HS. The study also highlighted the importance of the α-helical domain's electric charge and the influence of the Mg(II) ion in stabilizing this interaction. The work's impact on the field is significant, offering new perspectives for developing antiviral treatments for MPXV and potentially other viruses with similar adhesion mechanisms. The provided methods and data are highly useful for researchers working with viral proteins and protein-polysaccharide interactions, offering a solid foundation for future investigations and therapeutic innovations.

Comments on revised version:

The authors have successfully addressed the questions raised in my review.

---

## [Referee Report · Reviewer #2 (Public Review)]

Summary:

The manuscript presenting the discovery of a heparan-sulfate (HS) binding domain in monkeypox virus (MPXV) H3 protein as a new anti-poxviral drug target, presented by Bin Zhen and co-workers, is of interest, given that it offers a potentially broad antiviral substance to be used against poxviruses. Using new computational biology techniques, the authors identified a new alpha-helical domain in the H3 protein, which interacts with cell surface HS, and this domain seems to be crucial for H3-HS interaction. Given that this domain is conserved across orthopoxviruses, authors designed protein inhibitors. One of these inhibitors, AI-PoxBlock723, effectively disrupted the H3-HS interaction and inhibited infection with Monkeypox virus and Vaccinia virus. The presented data should be of interest to a diverse audience, given the possibility of an effective anti-poxviral drug.

Strengths:

In my opinion, the experiments done in this work were well-planned and executed. The authors put together several computational methods, to design poxvirus inhibitor molecules, and then they test these molecules for infection inhibition.

Comments on revised version:

The authors have addressed the comments I made in my review.

---

## [Referee Report · Reviewer #3 (Public Review)]

Summary:

The article is an interesting approach to determining the MPOX receptor using "in silico" tools. The results show the presence of two regions of the H3 protein with a high probability of being involved in the interaction with the HS cell receptor. However, the α-helical region seems to be the most probable, since modifications in this region affect the virus binding to the HS receptor.

Strengths:

In my opinion, it is an informative article with interesting results, generated by a combination of "in silico" and wet science to test the theoretical results. This is a strong point of the article.

Comments on revised version:

After a review of the changes to the manuscript and the author's responses, no further changes are needed.

---

## [Author Response]

The following is the authors’ response to the original reviews.

**Public Reviews:**

**Reviewer #1 (Public Review):**
Summary:The study aimed to better understand the role of the H3 protein of the Monkeypox virus (MPXV) in host cell adhesion, identifying a crucial α-helical domain for interaction with heparan sulfate (HS). Using a combination of advanced computational simulations and experimental validations, the authors discovered that this domain is essential for viral adhesion and potentially a new target for developing antiviral therapies.Strengths:The study's main strengths include the use of cutting-edge computational tools such as AlphaFold2 and molecular dynamics simulations, combined with robust experimental techniques like single-molecule force spectroscopy and flow cytometry. These methods provided a detailed and reliable view of the interactions between the H3 protein and HS. The study also highlighted the importance of the α-helical domain's electric charge and the influence of the Mg(II) ion in stabilizing this interaction. The work's impact on the field is significant, offering new perspectives for developing antiviral treatments for MPXV and potentially other viruses with similar adhesion mechanisms. The provided methods and data are highly useful for researchers working with viral proteins and protein-polysaccharide interactions, offering a solid foundation for future investigations and therapeutic innovations.Weaknesses:However, some limitations are notable. Despite the robust use of computational methodologies, the limitations of this approach are not discussed, such as potential sources of error, standard deviation rates, and known controls for the H3 protein to justify the claims. Additionally, validations with methodologies like X-ray crystallography would further benefit the visualization of the H3 and HS interaction.

Thank you very much for the evaluation and appreciation of our work. In response to the identified weakness, we have conducted additional analyses to further assess the limitations of the computational methodologies used. Specifically, we predicted the MPXV H3 structure using two other AI-based protein structure prediction models, ESMFold and RoseTTAFold2. Both models also predicted an a-helical structure, which supports our conclusion. However, they yielded lower pLDDT scores (Figure S1A-C in the revised SI), indicating that some error may be present.

We agree with this reviewer, as well as the other reviewers, that X-ray crystallography data for the H3 structure would be highly valuable. Unfortunately, we lack the expertise in structural biology to obtain these results at this stage. To complement this, we performed molecular dynamics (MD) simulations, which suggest that the helical domain is connected to the main domain via a flexible linker. This flexibility may help explain the challenges in obtaining a high-resolution X-ray structure. In fact, to date, the only structural data available for H3 is from the VAVC, which excludes the helical domain (The helical domain part is cleaved for the X-ray studies). We have added this point to the discussion and hope that experts in structural biology will be able to resolve the structure of this domain in the future.

**Reviewer #2 (Public Review):**
Summary:The manuscript presenting the discovery of a heparan-sulfate (HS) binding domain in monkeypox virus (MPXV) H3 protein as a new anti-poxviral drug target, presented by Bin Zhen and co-workers, is of interest, given that it offers a potentially broad antiviral substance to be used against poxviruses. Using new computational biology techniques, the authors identified a new alpha-helical domain in the H3 protein, which interacts with cell surface HS, and this domain seems to be crucial for H3-HS interaction. Given that this domain is conserved across orthopoxviruses, authors designed protein inhibitors. One of these inhibitors, AI-PoxBlock723, effectively disrupted the H3-HS interaction and inhibited infection with Monkeypox virus and Vaccinia virus. The presented data should be of interest to a diverse audience, given the possibility of an effective anti-poxviral drug.Strengths:In my opinion, the experiments done in this work were well-planned and executed. The authors put together several computational methods, to design poxvirus inhibitor molecules, and then they test these molecules for infection inhibition.Weaknesses:One thing that could be improved, is the presentation of results, to make them more easily understandable to readers, who may not be experts in protein modeling programs. For example, figures should be self-explanatory and understood on their own, without the need to revise text. Therefore, the figure legend should be more informative as to how the experiments were done.

Thank you very much for your appreciation of our work and your support. In response to the identified weakness, we have carefully reviewed all the figure legends to ensure they are more informative.

**Reviewer #3 (Public Review):**
Summary:The article is an interesting approach to determining the MPOX receptor using "in silico" tools. The results show the presence of two regions of the H3 protein with a high probability of being involved in the interaction with the HS cell receptor. However, the α-helical region seems to be the most probable, since modifications in this region affect the virus binding to the HS receptor.Strengths:In my opinion, it is an informative article with interesting results, generated by a combination of "in silico" and wet science to test the theoretical results. This is a strong point of the article.Weaknesses:Has a crystal structure of the H3 protein been reported?The following text is in line 104: "which may represent a novel binding site for HS". It is unclear whether this means this "new binding site" is an alternative site to an old one or whether it is the true binding site that had not been previously elucidated.

Thank you very much for your thoughtful evaluation and appreciation of our work.

We agree with this reviewer, as well as the other reviewers, that X-ray crystallography data for the H3 structure would be highly valuable. Unfortunately, we are not experts in structural biology, and we have not yet been able to obtain these structural results. To date, the only structure available for H3 is the one from VAVC, which does not include the helical domain. We have included this point in the discussion and hope that experts in structural biology will be able to resolve the structure of this domain in the future.

Regarding the "novel binding site," this term refers to "the true binding site that had not been previously elucidated." Previous research identified that H3 binds to heparan sulfate (HS), but the exact binding site had not been determined.

**Recommendations for the authors:**

**Reviewer #1 (Recommendations For The Authors):**
Validation of Results with Other Experimental Methods: While single-molecule force spectroscopy and flow cytometry provide valuable data, including complementary methods such as X-ray crystallography could offer additional insights into the H3-HS interaction and the effectiveness of the inhibitors.Discussion of Computational Model Limitations: Although the use of AlphaFold2 and other advanced tools is a strength, it is important to discuss the limitations of these models in more detail, including potential sources of error and how they may impact the interpretation of the results.During the manuscript evaluation, it is not clear the protein localization (transmembrane?) since the protein`s end is very close to the virus membrane surface. All experiments demonstrated the protein without being anchored to the membrane, letting the interaction site always be exposed. If the protein is linked to the membrane, how would the site be exposed due to the limited space between it and the virus structure?

Thank you for these insightful comments. As you pointed out, the H3 protein, particularly the helical domain at the C-terminal, is indeed located close to the membrane, which could limit the available space for H3 binding. To investigate this further, we modeled the full-length H3 protein in the context of the membrane and performed molecular dynamics (MD) simulations to assess the available space. Our results show that there is more than 1 nm of space between the helical domain and the membrane, which should be sufficient for potential heparan sulfate (HS) binding (see Figure 1E, and Figure S1D&E in the revised manuscript).

Minor corrections:Line 31: "is an emerging zoonotic pathogen" should be revised to reflect that Mpox is a re-emerging virus, given its history of causing outbreaks, such as in 2003.Line 71 and Line 75: Adding an explanation of "Mg binding sites" and "GAG motifs" would enhance reader understanding, as these represent important points in the study. The current positioning of Figure 1 causes some confusion for the reader.Line 111: High score? What controls were used for the protein? Are there known inhibitors of H3? If so, why weren't they tested for structure comparison? Additionally, what about other molecules that H3 binds to, such as UDP-Glucose, as demonstrated in the base article for the Vaccinia virus H3 protein available in the PDB?Figure 2B: Improve the legend, as the colors of the lines are not clear.

Thank you for your instructive comments. We have addressed most of them in the revised manuscript.

Regarding the "high score," AlphaFold2 provides a confidence score for its protein structure predictions, with a maximum score of 100. A score above 80 indicates a high level of confidence in the prediction.

There are known inhibitors (such as antibodies) of H3, and while the sequence is available, no structure has been reported so far. Previous s NMR titration measurements have shown that UDP-glucose binds to H3, but no structural data for the complex exist. To date, the only available crystal structure is of a truncated H3, which does not include the helical domain we identified from VAVC.

**Reviewer #2 (Recommendations For The Authors):**
The text described in the result section does not match the text presented in Figures. So, it is not easy to see what are the authors referring to when they mention the Figure. For example, the text referring to Figure S8 mentions the GB1 domain and the Cohesin module, but these are not mentioned in Figure S8.I do not understand the results presented in Figure 5B. It is not clear to me, from the Figure legend nor after reading the Material and Methods, how this experiment was done. Specifically, what is plotted on X, is it the amount of inhibitor or the amount of protein? These things have to be checked through the manuscript.It would be interesting to confirm if the inhibition of infection is based on the inhibition of viral binding to the cells. This should not be complicated to realize, and it could provide evidence for the mechanism of action.Extensive use of terms like "this domain" is not good in this type of article, like in lines 207, and 211. It is not always clear to what domain are authors referring to, so it may be much better to mention the domain in question by the exact name.Line 337, If I am not mistaken dilutions are serial not series.Line 613, in methods. Please use g force instead of rpm, it is more informative. Even if it is just to pellet cells.

Thank you very much for your instructive comments. We have addressed most of them in the revised manuscript. For instance, the immobilization of the GB1 domain and the cohesin module is now mentioned in Figure S9. Additionally, in the previous Figure 5B, the "x" represents the concentration of the inhibitor. Serial and g force is updated.

**Reviewer #3 (Recommendations For The Authors):**
Line 190Did you mutate all the amino acids at the same time? What was the impact of all these mutations on the structure of the helical region? Or if you modeled the protein again after replacing these 7 amino acids, did you find that there was no difference? Regardless of your answer, you must include a superposition of the mutated structure and the wt.

Thank you for the insightful comment. We have now also predicted the structure of the serine mutant using AlphaFold2 (AF2). As expected, the helical domain structure remains largely preserved with only minor differences. We have included these results in Figure S6, as suggested.

Figure 2DIn this graph, the authors should indicate the ΔG as a negative value. In fact, the graph does not match the text.

Thanks for the reminder, it is corrected in the graph

Figure 4BIs the difference in binding force significantly different? 28.8 vs 33.7 pN

The absolute difference in binding force is not large (~5 pN). However, for a system with a relatively low binding force, this difference is significant. Specifically, the 5 pN difference accounts for approximately a 14% reduction in binding force. We have included this percentage in the revised manuscript.

Figure 5If AI-PoxBlocks723 was the only peptide effective in inhibiting viral infection of MPOX and other related viruses but not with 100% effectiveness, do you think this could be a consequence of a low interaction efficiency or the existence of a different receptor? Or a secondary region of binding in the H3? Can you argue about this?

It has been proposed that there are other adhesion proteins for MPXV, such as D8, in addition to H3. We believe this accounts for the observed less-than-100% effectiveness.

The use of peptides as "inhibitory tools" could have an interesting effect in vitro, however, in vivo the immunological response against the peptide will reduce/eliminate it, how you may optimize the "drug" development with this system, as you state in line 387.

Thank you for your thoughtful comment. You are correct that the use of peptides as inhibitory tools could induce an immune response in vivo, which might limit their effectiveness over time. To optimize this approach for drug development, conjugate the peptides with carrier molecules, such as liposomes, nanoparticles, or dendrimers, which can protect the peptides from immune detection and improve their delivery to target cells. This could allow for more controlled and sustained release of the peptide in vivo, reducing the chances of immune clearance. We have added this discussion in the revised manuscript.